# Dependence of turbulence estimations on nacelle-lidar scanning strategies

Wei Fu, Alessandro Sebastiani, Alfredo Peña, and Jakob Mann

Department of Wind and Energy Systems, Technical University of Denmark, Frederiksborgvej 399, 4000 Roskilde, Denmark

**Correspondence:** Wei Fu (weif@dtu.dk)

**Abstract.** Through numerical simulations and the analysis of field measurements, we investigate the dependence of the accuracy and uncertainty of turbulence estimations on the main features of the nacelle lidars' scanning strategy, i.e., the number of measurement points, the half-cone opening angle, the focus distance and the type of the lidar system. We assume homogeneous turbulence over the lidar scanning area in front of a Vestas V52 wind turbine. The Reynolds stresses are computed via a least-squares procedure that uses the radial velocity variances of each lidar beam without the need to reconstruct the wind components. The lidar-retrieved Reynolds stresses are compared with those from a sonic anemometer at turbine hub height. Our findings from the analysis of both simulations and measurements demonstrate that to estimate the six Reynolds stresses accurately, a nacelle lidar system with at least six beams is required. Further, one of the beams of this system should have a different opening angle. Adding one central beam improves the estimations of the velocity components' variances. Assuming the relations of the velocity components' variances as suggested in the IEC standard, all considered lidars can estimate the along-wind variance accurately using the least-squares procedure and the Doppler radial velocity spectra. Increasing the opening angle increases the accuracy and reduces the uncertainty on the transverse components, while enlarging the measurement distance has opposite effects. All in all, a 6-beam continuous-wave lidar measuring at a close distance with a large opening angle provides the best estimations of all Reynolds stresses. This work gives insights on designing and utilizing nacelle lidars for inflow turbulence characterization.

## 1 Introduction

Inflow turbulence characteristics are important for wind turbine load validation (Conti et al., 2021), power performance assessment (Gottschall and Peinke, 2008; Wagner et al., 2014) and wind turbine control (Dong et al., 2021). The traditional way to measure inflow turbulence uses the in-situ anemometers installed on meteorological masts, such as cup and sonic anemometers. However, rotor planes of the modern wind turbines have large vertical span that can reach 250 m above the ground. It is more and more costly to install a meteorological mast that reaches the height of the blade tips, especially under offshore conditions. Recently, nacelle lidars of different types and configurations have been used to scan the inflow (Harris et al., 2006; Mikkelsen et al., 2013; Wagner et al., 2015; Peña et al., 2017; Fu et al., 2022a). Compared to the point-wise, mast-mounted anemometers, forward-looking nacelle lidars yaw with the wind turbine and measure at different points in front of the rotor, which can potentially better characterize the inflow that actually interacts with the wind turbine.

Assuming statistical stationarity, turbulence can be represented by the variances and covariances of the wind field components $u, v$ and $w$ ($u_1, u_2, u_3$) averaged typically over 10 or 30 min. The homogeneous velocity field can be decomposed into the mean $U_i$ and the fluctuating part $u_i'$. The Reynolds stress tensor, a matrix containing the six second-order moments $\langle u_i' u_j' \rangle$, describes the variability of the atmospheric flow in some detail. The terms in the Reynolds stress tensor are frequently used in wind energy and meteorology. The square root of the along-wind variance is a part of the definition in the turbulence intensity, which is a key turbulence parameter for the structural loads assessment and the design of wind turbines (IEC, 2019). However, this is not the only component that is important for loads (Petersen et al., 1994). The two covariances $\langle u'w' \rangle$ and $\langle v'w' \rangle$ form the momentum fluxes, which are used to calculate the friction velocity and are closely connected to the vertical wind profile (Wyngaard, 2010; Peña et al., 2016). The half the sum of the variances of the three velocity components is the turbulence kinetic energy, which is an important parameter for investigating wind turbine wake structures (Kumer et al., 2016). Also, the Reynolds stresses are needed to determine the parameters of the three-dimensional turbulence models for, e.g., load simulations (Mann, 1994).

Compared to turbulence estimates from traditional anemometry, the accuracy and the uncertainty of lidar-derived turbulence characteristics can be affected by not only the spatial and temporal resolutions intrinsic to the lidar systems and the characteristics of atmospheric turbulence but also the lidar scanning strategies (Sathe et al., 2011; Smalikho and Banakh, 2017). Dimitrov and Natarajan (2017) and Conti et al. (2021) applied lidar measurements using different scanning strategies for load validation. Schlipf et al. (2018) optimized the scanning trajectory of nacelle lidars based on a coherence model for the rotor-effective wind speed to improve control performance. Only a few works investigated the dependence of turbulence estimations on lidar scanning strategies. Sathe et al. (2015) explained that at least six radial velocity variances are needed to compute all six Reynolds stresses, and proposed for a ground-based lidar an optimized six-beam configuration using an objective function which minimizes the sum of the random errors of the Reynolds stresses. Newman et al. (2016) showed that using the variance from the vertical beam improves the turbulence estimates from ground-based lidars. Fu et al. (2022a) investigated the benefit of using multiple-beam nacelle lidars by comparing the accuracy of turbulence estimations from a SpinnerLidar (a lidar measuring the inflow at 400 positions) with two- and four-beam lidars.

Lidars measure the radial velocity (also known as the line-of-sight velocity) along the laser beam. Sathe and Mann (2013) and Fu et al. (2022a) showed that the variance along a single beam can be higher or lower than the $u$-variance measured by sonic anemometers depending on the beam orientation. This is due to the correlation between different velocity components, which can be described in the three-dimensional spectral velocity tensor model by Mann (1994) (hereafter Mann model). We need to assume homogeneity when combining the radial velocity variances along different laser beam directions to reconstruct the Reynolds stresses. Compared to the in-situ anemometers, the lidar's measurement volume is generally larger, which leads to turbulence attenuation.

There are two main types of nacelle lidar systems, namely continuous-wave (CW) and pulsed. They mainly differ on the working principle and the way they probe the atmosphere within their measurement volume. The probe volume of a CW system increases with the square of the focus distance, while the one of a pulsed system remains constant with measurement range (Peña et al., 2015). The 'unfiltered' radial velocity variances (in which the volume-averaging effect is compensated) can be

retrieved from the Doppler radial velocity spectra, which are normally available in CW systems (Mann et al., 2010; Branlard et al., 2013).

This work investigates the dependence of the accuracy and the uncertainty of the turbulence estimations on the main features of the nacelle lidars' scanning strategy, i.e., the number of measurement positions within a full scan, the half-cone opening angle, the focus distance and the type of the lidar system. We select eight scanning patterns, which are commonly known or widely used in the wind energy industry. Homogeneous frozen turbulence is assumed throughout our analysis. The Reynolds stresses are estimated via a least-squares procedure using radial velocity variances instead of computing from the reconstructed mean wind velocities. Estimates from a sonic anemometer at turbine hub height are used as reference. Compared to Fu et al. (2022b), here we study the topic using not only numerical simulations with turbulence boxes but also the SpinnerLidar measurements collected at DTU Risø test site. We select measurements at certain beam scanning locations of the SpinnerLidar to imitate lidars with different scanning configurations. Another main difference to Fu et al. (2022b) is that we consider the probe volume of both a CW and a pulsed lidar system in our simulations, which plays an important role, especially when studying the influence of the focus distance on the turbulence estimation.

This paper is organized as follows. Section 2 introduces the turbulence spectral model and the modeling of nacelle lidars. Section 3 describes how the unfiltered radial velocity variance and the Reynolds stresses are estimated. It also gives details about the setup of the numerical simulations, the considered lidar scanning strategies and the field experiment. Section 4 compares the Reynolds stress estimations between the lidars and the sonic anemometer at turbine hub height from both numerical simulations and measurements. Discussions are given in Section 5. Section 6 concludes the work and provides the outlook.

## 2 Theoretical background

### 2.1 Turbulence spectral model

Assuming Taylor's frozen turbulence (Taylor, 1938), the wind field can be described by $\boldsymbol{u}(\boldsymbol{x}) = (u, v, w)$, where $\boldsymbol{x} = (x, y, z)$ is the position vector defined in a right-handed coordinate system, $u$ the horizontal along-wind component, $v$ the horizontal lateral component, and $w$ the vertical component. The homogeneous wind field $\boldsymbol{u}(\boldsymbol{x})$ can be decomposed into the mean value $\langle \boldsymbol{u}(\boldsymbol{x}) \rangle = (U, 0, 0)$, where $\langle \ \rangle$ denotes ensemble averaging, and the fluctuating part $\boldsymbol{u}'(\boldsymbol{x}) = (u', v', w')$. $U$ is the mean wind velocity along the $x$-direction. The one-dimensional single point (co-)spectra of any component of the wind field are given as (Mann, 1994)

$$F_{ij}(k_1) = \frac{1}{(2\pi)} \int\limits_{-\infty}^{\infty} R_{ij}(x_1, 0, 0)\exp(-ik_1 \cdot x_1)\mathrm{d}x_1, \tag{1}$$

where $k_1$ is the first component of the wave vector $\boldsymbol{k}$, $R_{ij}(\boldsymbol{r}) \equiv \langle u_i'(\boldsymbol{x})u_j'(\boldsymbol{x}+\boldsymbol{r}) \rangle$ is the Reynolds stress tensor, $\boldsymbol{r}$ is the separation vector, and $u_i'$ are the fluctuations around the mean of the wind field. The wave number can, via Taylor's hypothesis, be related to the frequency $f$ through $k_1 = 2\pi f/U$. The auto-spectra of the three wind components $F_{u,v,w}$ ($= F_{11,22,33}$) can

be evaluated using Eq. (1). The velocity components' variances are

$$\sigma^2_{u,v,w} = \int_{-\infty}^{\infty} F_{u,v,w}(k_1) \mathrm{d}k_1. \tag{2}$$

We assume that the Mann model well describes the spatial structure of the turbulent flow. Besides $k_1$ and the other two components of the wave vector $\boldsymbol{k}$, the Mann model contains three parameters: $\alpha\varepsilon^{2/3}$, which is related to the turbulent energy dissipation rate, $L$ to a turbulence length scale, and $\Gamma$ to the anisotropy of turbulence. This model is chosen because it describes the correlations between different velocity components, which play an important role in deriving turbulence statistics from measurements of multiple-beam lidars pointing at different directions.

## 2.2 Nacelle lidar and modeling of the probe volume

The unit vector $\boldsymbol{n}$ describes the beam orientation of a nacelle lidar, which can be expressed as (Peña et al., 2017):

$$\boldsymbol{n}(\phi,\theta) = (-\cos\phi, \cos\theta\sin\phi, \sin\theta\sin\phi), \tag{3}$$

where $\theta$ is the angle between the $y$ axis and $\boldsymbol{n}$ projected onto the $y$-$z$ plane and $\phi$ the angle between the beam and the negative $x$-axis (also known as the half-cone opening angle), as shown in Fig.1.

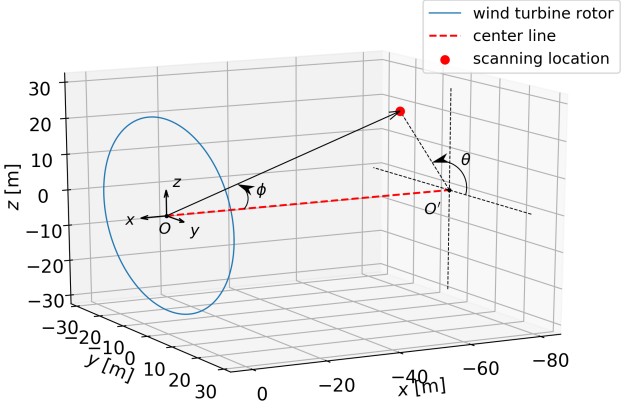

**Figure 1.** Definition of the coordinate system and beam angles for nacelle lidar modeling.

The radial velocity of a lidar can be written as the convolution of the weighting function $\varphi$ and the radial velocity sampled along the beam in the probe volume (Mann et al., 2010):

$$v_\mathrm{r}(\phi,\theta) = \int_{-\infty}^{\infty} \varphi(s)\boldsymbol{n}(\phi,\theta) \cdot \boldsymbol{u}[\boldsymbol{n}(\phi,\theta)(f_d+s)]\mathrm{d}s, \tag{4}$$

where $s$ is the distance from the focus point along the beam and $f_d$ the focus or measurement distance. The relation assumes that the velocity is determined from the Doppler spectrum as the center of gravity, see Held and Mann (2018). We use the following weighting functions to approximate the probe volume of different types of lidar:

- CW lidar (Sonnenschein and Horrigan, 1971):

$$\varphi(s) = \frac{1}{\pi} \frac{z_{\mathrm{R}}}{z_{\mathrm{R}}^2 + s^2} \quad \text{with } z_{\mathrm{R}} = \frac{\lambda f_d^2}{\pi r_{\mathrm{b}}^2}, \tag{5}$$

where $z_{\mathrm{R}}$ is the Rayleigh length, $\lambda$ the laser wavelength and $r_{\mathrm{b}}$ the beam radius at the output lens.

- pulsed lidar (Meyer Forsting et al., 2017):

$$\varphi(s) = \frac{1}{2\Delta p} \left\{ \mathrm{Erf}\left[\frac{s + \Delta p/2}{r_p}\right] - \mathrm{Erf}\left[\frac{s - \Delta p/2}{r_p}\right] \right\}$$

$$\text{with the error function } \mathrm{Erf}(x) = \frac{2}{\sqrt{\pi}} \int_0^x \exp(-t^2)\mathrm{d}t \quad \text{and} \quad r_p = \frac{\Delta l}{2\sqrt{\ln(2)}}, \tag{6}$$

where $\Delta p$ is the range-gate length and $\Delta l$ the Gaussian lidar pulse Full Width at Half Maximum (FWHM).

Variances calculated from the centroid-derived radial velocities are attenuated by the lidar probe volume, which acts like a low-pass filter to the wind velocity fluctuations. Therefore, we refer to them as the 'filtered' radial velocity variances. If we assume that the lidar probe volume can be negligible and that $u, v$, and $w$ are constant over the scanned area, the radial velocity can be expressed as

$$v_{\mathrm{r}}(\phi, \theta) = -u\cos\phi + v\cos\theta\sin\phi + w\sin\theta\sin\phi. \tag{7}$$

The 'unfiltered' radial velocity variance can be derived by taking the variance of Eq. (7), as shown in Eberhard et al. (1989):

$$\sigma_{v_{\mathrm{r}},\mathrm{unf}}^2(\phi, \theta) = \sigma_u^2\cos^2\phi + \sigma_v^2\cos^2\theta\sin^2\phi + \sigma_w^2\sin^2\theta\sin^2\phi - 2\langle u'v'\rangle\cos\phi\cos\theta\sin\phi$$

$$- 2\langle u'w'\rangle\cos\phi\sin\theta\sin\phi + 2\langle v'w'\rangle\sin^2\phi\cos\theta\sin\theta. \tag{8}$$

## 3 Methodology

### 3.1 Estimation of the unfiltered radial velocity variance

In practice, the unfiltered radial velocity variance $\sigma_{v_{\mathrm{r}},\mathrm{unf}}^2$ in Eq. (8) can be estimated from the Doppler radial velocity spectrum. When the nacelle lidar measures at a small opening angle over a relatively homogeneous inflow and the wind shear is not very strong, the effect of radial velocity gradient within the lidar probe volume can be negligible (see Mann et al., 2010, for a detailed discussion). In this case, one can estimate $\sigma_{v_{\mathrm{r}},\mathrm{unf}}^2$ as the second central statistical moment of the ensemble-averaged Doppler spectrum of the radial velocity within typically a 10- or 30-min period. Each Doppler spectrum is area-normalized

before computing the ensemble-averaged Doppler spectrum $p(v_\mathrm{r})$. The mean radial velocity can be estimated as

$$\mu_{v_\mathrm{r}} = \int_{-\infty}^{\infty} v_\mathrm{r} p(v_\mathrm{r}) \mathrm{d}v_\mathrm{r}, \tag{9}$$

and its variance as

$$\sigma_{v_\mathrm{r}}^2 = \int_{-\infty}^{\infty} (v_\mathrm{r} - \mu_{v_\mathrm{r}})^2 p(v_\mathrm{r}) \mathrm{d}v_\mathrm{r}. \tag{10}$$

Assuming that all contributions of the radial velocity to the Doppler spectrum are because of turbulence, $\sigma_{v_\mathrm{r}}^2$ in Eq. (10) provides an estimate of $\sigma_{v_\mathrm{r},\mathrm{unf}}^2$. This assumption is reasonable when beams are close to horizontal.

## 3.2 Estimation of the Reynolds stresses

The Reynolds stress tensor $\boldsymbol{R} \equiv \boldsymbol{R}(\boldsymbol{x} = \boldsymbol{0})$ contains the variances and covariances of the velocity components:

$$\boldsymbol{R} = \begin{bmatrix} \sigma_u^2 & \langle u'v' \rangle & \langle u'w' \rangle \\ \langle v'u' \rangle & \sigma_v^2 & \langle v'w' \rangle \\ \langle w'u' \rangle & \langle w'v' \rangle & \sigma_w^2 \end{bmatrix}. \tag{11}$$

To compute $\boldsymbol{R}$, we use the radial velocity variances from all beams over the lidar scanning trajectory. Assuming spatial homogeneity, we apply a least-squares fit to the radial velocity variances $\sigma_{v_\mathrm{r}}^2$. This can be done since the variance in any direction $\boldsymbol{n}$ can be written as $\boldsymbol{n} \cdot \boldsymbol{R}\boldsymbol{n}$ or $n_i R_{ij} n_j$ using the index notation and assuming summation over repeated indices. We then sum the squared differences between the measured radial variances $\sigma_{v_\mathrm{r}}^2$ and $\boldsymbol{n} \cdot \boldsymbol{R}\boldsymbol{n}$ for any given Reynolds stress tensor $\boldsymbol{R}$. In order to avoid too many indices, we express this sum as integral $\int \mathrm{d}\mu$ such that the sum we are going to minimize can be written as

$$\Delta^2 = \int (\boldsymbol{n} \cdot \boldsymbol{R}\boldsymbol{n} - \sigma_{v_\mathrm{r}}^2)^2 \mathrm{d}\mu. \tag{12}$$

The matrix $R_{ij}$ that minimizes the integral must fulfill

$$\frac{\partial \Delta^2}{\partial R_{ij}} = 0 \Rightarrow 2 \int (\boldsymbol{n} \cdot \boldsymbol{R}\boldsymbol{n} - \sigma_{v_\mathrm{r}}^2) n_i n_j \mathrm{d}\mu = 0. \tag{13}$$

This can be written as

$$R_{kl} \int n_k n_l n_i n_j \mathrm{d}\mu = \int \sigma_{v_\mathrm{r}}^2 n_i n_j \mathrm{d}\mu, \tag{14}$$

where $(k,l)$ and $(i,j)$ are each of the indices combinations $(1,1),(1,2),(1,3),(2,2),(2,3),(3,3)$, $n_1 = -\cos\phi$, $n_2 = \cos\theta\sin\phi$ and $n_3 = \sin\theta\sin\phi$ (as given in Eq. 3), i.e. Fu et al. (2022a),:

$$\begin{bmatrix} \sum n_1^4 & \sum n_1^2 n_2^2 & \sum n_1^2 n_3^2 & \sum 2n_1^3 n_2 & \sum 2n_1^3 n_3 & \sum 2n_1^2 n_2 n_3 \\ \sum n_1^2 n_2^2 & \sum n_2^4 & \sum n_2^2 n_3^2 & \sum 2n_1 n_2^3 & \sum 2n_1 n_2^2 n_3 & \sum 2n_2^3 n_3 \\ \sum n_1^2 n_3^2 & \sum n_2^2 n_3^2 & \sum n_3^4 & \sum 2n_1 n_2 n_3^2 & \sum 2n_1 n_3^3 & \sum 2n_2 n_3^3 \\ \sum n_1^3 n_2 & \sum n_1 n_2^3 & \sum n_1 n_2 n_3^2 & \sum 2n_1^2 n_2^2 & \sum 2n_1^2 n_2 n_3 & \sum 2n_1 n_2^2 n_3 \\ \sum n_1^3 n_3 & \sum n_1 n_2^2 n_3 & \sum n_1 n_3^3 & \sum 2n_1^2 n_2 n_3 & \sum 2n_1^2 n_3^2 & \sum 2n_1 n_2 n_3^2 \\ \sum n_1^2 n_2 n_3 & \sum n_2^3 n_3 & \sum n_2 n_3^3 & \sum 2n_1 n_2^2 n_3 & \sum 2n_1 n_2 n_3^2 & \sum 2n_2^2 n_3^2 \end{bmatrix} \begin{bmatrix} R_{uu} \\ R_{vv} \\ R_{ww} \\ R_{uv} \\ R_{uw} \\ R_{vw} \end{bmatrix} = \begin{bmatrix} \sum n_1^2 \sigma_{v_\mathrm{r}}^2 \\ \sum n_2^2 \sigma_{v_\mathrm{r}}^2 \\ \sum n_3^2 \sigma_{v_\mathrm{r}}^2 \\ \sum n_1 n_2 \sigma_{v_\mathrm{r}}^2 \\ \sum n_1 n_3 \sigma_{v_\mathrm{r}}^2 \\ \sum n_2 n_3 \sigma_{v_\mathrm{r}}^2 \end{bmatrix}. \tag{15}$$

To solve the six Reynolds stresses from Eq. (15), two requirements of the nacelle lidar scanning pattern need to be fulfilled (see Sathe et al., 2015, for a detailed discussion):

- the lidar has at least six beams or measures at six different locations within one full scan;

- the lidar beams have at least two different opening angles.

If a lidar has less than six beams, or the opening angles of all beams are identical and some of the six equations are linearly dependent, we have fewer knowns than unknowns in Eq. (15), which leads to infinite solutions. In those cases, only the along-wind variance $\sigma_u^2$ can be estimated well (Peña et al., 2019). To solve $\sigma_u^2$ from Eq. (15), assumptions of some Reynolds stresses terms are needed to reduce the number of unknowns. Here, we use three different assumptions, as introduced in Fu et al. (2022a):

- All Reynolds stresses apart from $\sigma_u^2$ are zero (denoted as 'LSP-$\sigma_u^2$' method). For lidars with only one half-cone opening angle, this means $\sigma_u^2 = \sum \sigma_{v_r}^2 / \sum \cos^2 \phi$.

- Turbulence is isotropic, i.e., $\sigma_u^2 = \sigma_v^2 = \sigma_w^2$ and that other terms are negligible (denoted as 'LSP-isotropy' method). This method is the same for lidars with only one half-cone opening angle as taking the mean of all radial velocity variances.

- The relations between velocity components' standard deviation $\sigma_v = 0.7\sigma_u$ and $\sigma_w = 0.5\sigma_u$, as recommended in IEC (2019), and all covariances are negligible (denoted as 'LSP-IEC' method).

## 3.3 Numerical simulations

We simulate lidar measurements on the nacelle of a wind turbine with a rotor diameter ($D$) of 52 m using 100 randomly generated turbulence boxes. The boxes contain the fluctuations of the three wind components. The turbulence boxes are described by the Mann model with typical values of the model parameters $\alpha\varepsilon^{2/3} = 0.05$ m$^{4/3}$ s$^{-1}$, $L = 61$ m and $\Gamma = 3.2$. The selected three parameters are adopted from Mann (1994) and characterize a neutral atmospheric stratification on a typical offshore site. The dissipation rate $\alpha\varepsilon^{2/3}$ is a scaling factor on the turbulence intensity. The number of grid points in the three directions are $(N_x, N_y, N_z) = (8192, 64, 64)$. The lengths of the turbulence boxes in the vertical and lateral directions are both 128 m. The boxes have lengths of 30 min in the along-wind direction assuming a mean wind $U = 10$ m s$^{-1}$. We add a linear shear $dU/dz = 0.0288$ s$^{-1}$ on top of the along-wind velocity component $u$ in each box:

$$u = U + \frac{dU}{dz}(z - z_{\text{rotor}}) + u',$$ (16)

where $z_{\text{rotor}}$ is the turbine hub height in the turbulence box, i.e., the middle grid point in the $z$-coordinate.

We simulate eight lidars with different scanning patterns, as shown in Fig. 2. Statistics of the sonic anemometer are taken at the location of the turbine rotor center (which is also the center of the turbulence boxes) as the reference for evaluating the lidar-derived turbulence characteristics. The SpinnerLidar scans in a rosette-curve pattern and has half-cone opening angles in the range $0 - 30°$. It generates 400 radial velocities in one full scan. The SpinnerLidar is simulated with a focus distance of

52 m ($1D$) in front of the rotor, while other lidars are simulated with the focus distance of 98 m due to their smaller opening angles ($\phi = 15°$) to cover the whole rotor plane. We also simulate all considered lidars with multiple measurement planes at $f_d = 49$, 72, 98, 121 and 142 m, which are arbitrarily selected. As examples, Fig. 3 shows the scanning trajectories of the 4-beam and 50-beam lidars measuring at the five planes. We then use the radial velocity variances at all measurement levels to compute the turbulence statistics. Furthermore, to study the dependence of the turbulence estimations on the opening angle and the focus distance, we simulate the 6-beam configuration, proposed by Sathe et al. (2015), with extra setups: a fixed focus distance of 52 m and increasing opening angles (see Fig. 4(a)), as well as a fixed opening angle of $15°$ and increasing focus distances (see Fig. 4(b)).

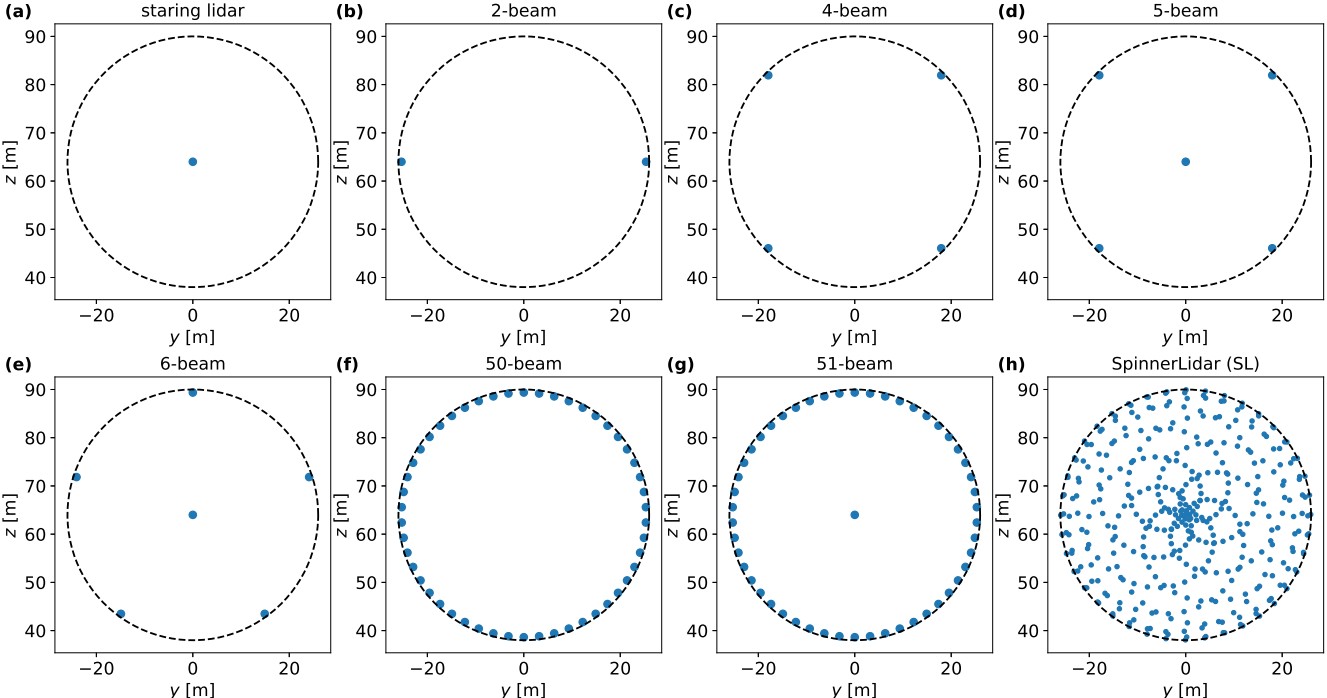

**Figure 2.** Selected lidar scanning patterns for numerical simulations. The SpinnerLidar (h) has $\phi = 0 - 30°$ and scans at $f_d = 52$ m, while other lidars (a-g) have $\phi = 15°$ and scan at $f_d = 98$ m to cover the whole rotor plane. The lidar beam scanning locations are marked in blue dots. The wind turbine rotor is represented in a black dashed circle.

We consider the lidar probe volume when we investigate the dependence of the Reynolds stresses estimation on $\phi$ and $f_d$. The Doppler radial velocity spectrum $S(v_r, t)$ is simulated as (Held and Mann, 2018)

$$S(v_r, t) = \int_{-M}^{M} \varphi(s)\delta(v_r - \boldsymbol{u}(\boldsymbol{n}s - \boldsymbol{U}t) \cdot \boldsymbol{n})\mathrm{d}s, \tag{17}$$

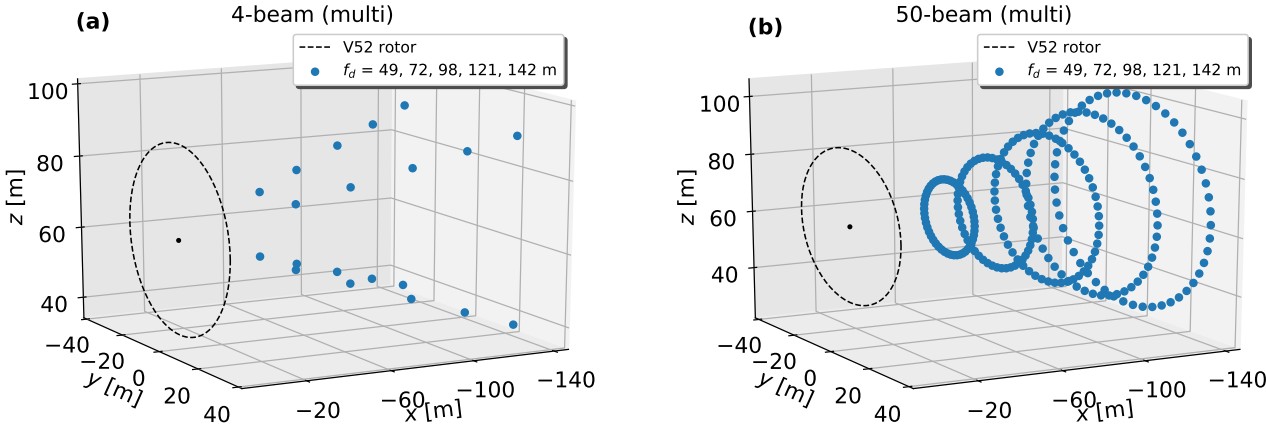

**Figure 3.** Scanning trajectories of the 4-beam and the 50-beam lidars measuring at $f_d = 49, 72, 98, 121$ and $142$ m. Features regarding the blue dots and the dashed circle as in Fig. 2. The turbine nacelle is marked in a black dot on the rotor plane.

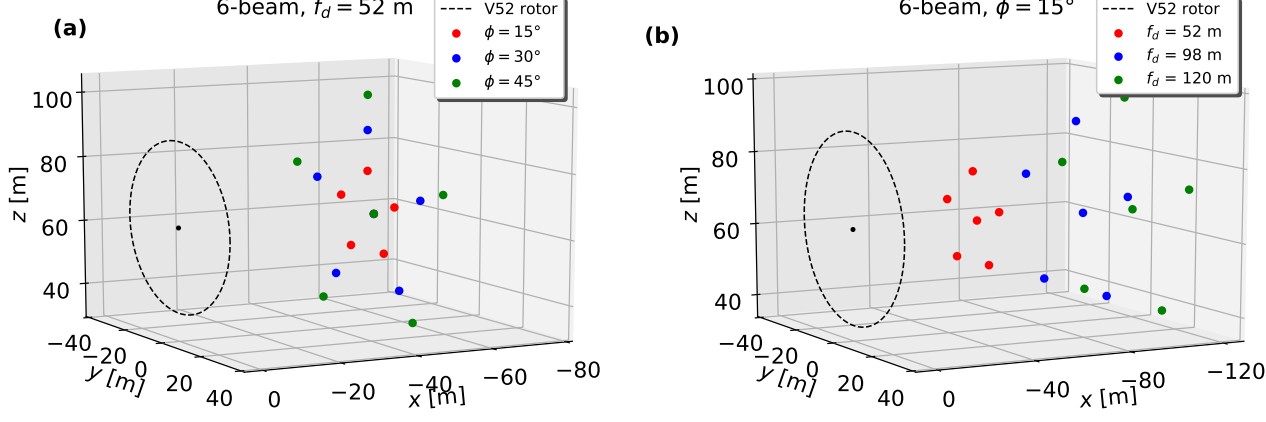

**Figure 4.** Scanning strategies of the 6-beam lidar with (a) a fixed focus distance and various half-cone opening angles, and (b) a fixed half-cone opening angle and various focus distances. Features regarding the dashed circle and the black dot as in Fig.3.

where $\delta$ represents the Dirac delta function, the integral is truncated with the distance $M$ along the beam, and $\varphi(s)$ can be described by Eq. (5) or (6) depending on the type of the lidar system. The resolution of the Doppler radial velocity spectrum is $0.1 \text{ m s}^{-1}$ per velocity bin, which is hereafter always used. Parameters used for modelling the probe volume are summarized in Table 1 (Meyer Forsting et al., 2017). We select $M$ as shown in Table 1 so that $95\%$ of the area under both weighting functions is covered. Figure 5 compares the modelled lidar probe volume for CW and pulsed lidars at focus distances $f_d = 52,\ 98$ and 120 m. The size of the probe volume for CW lidars increases with the square of the focus distance (see Eq. 5), while it remains the same for pulsed lidars.

| | |
|---|---|
| | $\lambda = 1.565 \times 10^{-6} \text{m}$ |
| CW | $r_b = 2.8 \times 10^{-2} \text{ m}$ |
| | $M = 8 z_{\text{R}}$ |
| | $\Delta l = 24.75 \text{ m}$ |
| pulsed | $\Delta p = 38.4 \text{ m}$ |
| | $M = 1.2 \Delta l$ |

**Table 1.** Parameters for modelling the CW and pulsed lidar probe volume in numerical simulations.

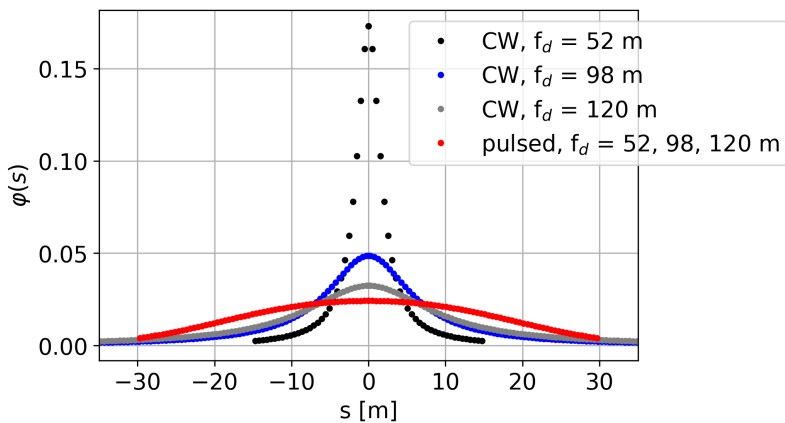

**Figure 5.** Comparison of the modelled lidar probe volume for CW and pulsed lidars at three different focus distances.

The time lag between each measurement within a full scan is not considered but assumed that measurements are taken at the same time. In the numerical simulations neglecting lidar probe volume (see results in Sections 4.1 and 4.2), the time resolution of the wind field is used as the lidar scan rate, i.e., lidars complete one full scan in $\text{d}t = \text{d}x/U = 0.22$ s. In the simulations considering lidar probe volume (see results in Section 4.3), the lidars are assumed to finish a full scan in 2 s.

### 3.4 Field measurements

During the period from 1 October 2020 to 30 April 2021, a SpinnerLidar was deployed on the nacelle of a Vestas V52 wind
turbine at DTU Risø campus in Roskilde, Denmark, measuring the flow in front of the turbine. The V52 wind turbine has a
rotor diameter of 52 m and a hub height of 44 m. Between the scan head of the SpinnerLidar and the turbine rotation axis, there
is a vertical displacement of 2.47 m. A test site layout is shown on a digital surface elevation model in Fig. 6. The terrain is
slightly hilly and its surface is characterized by a mix of cropland, grassland and coast. The dominant wind directions during
this period at this site are west and south-west. The V52 wind turbine (marked with a red circle) stands at the northernmost
position of a row of wind turbines (marked in black circles). There is also a meteorological mast (marked as a red square)
mounted at 120 m ($\approx 2.3D$) upstream from the V52 wind turbine at 291° from the north. One of the Metek USA-1 3D sonic
anemometers on the mast is located at 44 m above the ground, and its turbulence statistics is used as references to be compared
with the estimations from the nacelle-based lidars. A cup anemometer is located at the same height as the sonic anemometer
on the mast. There are also a wind vane at 41 m and a Thies precipitation opto sensor at 2 m on the mast.

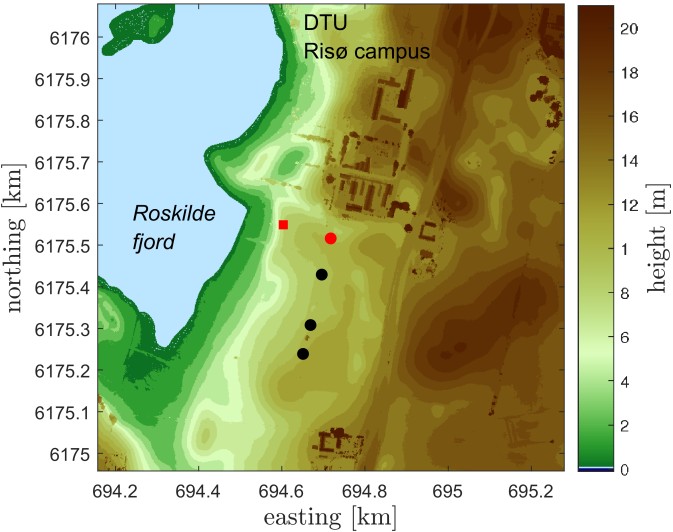

**Figure 6.** A digital surface elevation model (UTM32 WGS84) showing the Risø test site in Roskilde, Denmark. The height above the mean
sea level is indicated by the color bar (in meters). A row of wind turbines are marked in circles (in red the reference V52 wind turbine). The
meteorological mast is shown in a red square.

The SpinnerLidar (Peña et al., 2019) is based on a CW system and it was set up to scan the inflow at a focus distance of 62
m ($\approx 1.2D$, see Fig. 7). The Rayleigh length $z_R$ of the SpinnerLidar at this focused distance is 2.44 m. It reported 400 radial
velocities at a rate of 200 Hz, so it took 2 s to finish one full scan. The system also stored the instantaneous Doppler spectrum
of the radial velocity, which allows us to estimate the unfiltered radial velocity variance.

The measurements used for the analysis are from the wind sectors, which are relatively aligned with the mast-turbine direc-
tion (i.e., the 10-min averaged wind direction measured by the vane is within $291° \pm 30°$). The yaw misalignment of the V52

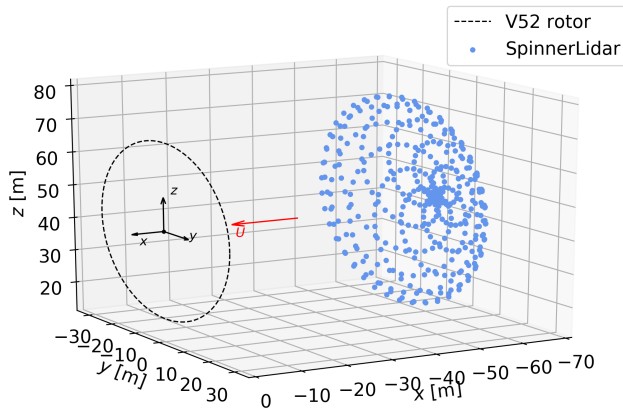

**Figure 7.** The scanning trajectory of the SpinnerLidar in the measurement campaign.

turbine is below $5°$, thereby minimizing the influence of nearby wind turbine wakes. We use a 10-min period, when the lidar and the V52 wind turbine are concurrently operating, and the averaged wind speed from the cup anemometer at $44$ m is higher than $3$ m s$^{-1}$. No precipitation was detected during the analyzed 10-min periods. After filtering, 2348 10-min periods are used for the analysis.

The SpinnerLidar measurements are post-processed to remove the signals reflected by the wind turbine blades, the telescope lens (the beam can hit the lens perpendicularly) or other hard targets. Such a procedure filters out some measurements close to the middle of the pattern. To compensate for the nacelle movement, we rotate the system-reported beam scanning coordinates using the 10-min averaged azimuthal and inclination angles of the SpinnerLidar, which are typically around $0.3°$ and $3°$, respectively. Taking the motion of the turbine and the slack of the SpinnerLidar into consideration, we divide the $y$–$z$ plane

into grids of 1-m resolution to aggregate the corrected scan locations. In the given 10-min, all Doppler radial velocity spectra lying within each grid cell are accumulated, and only measurements within the grid cells, where there are more than 30 instantaneous Doppler spectra, are used for the reconstruction. At least 900 grid cells should satisfy the criterion in the 10-min periods for our analysis. The light-grey dots in Figs. 8 and 9 represent the grid cells (for this particular case we have 1127 grid cells) satisfying the criterion in one arbitrary 10-min period. Other details about the measurement campaign and how the

SpinnerLidar measurements are selected, filtered and processed can be found in Fu et al. (2022a). The post-processing of the measurements leaves us 1294 time periods for the final comparison.

    To imitate lidars with different scanning strategies, we select SpinnerLidar measurements at certain grid cells to estimate the Reynolds stresses, as marked in red in Fig. 8. Due to the rotation of the system-reported lidar unit vectors, the corresponding half-cone opening angles of the grid cells are typically higher in the upper circle than those in the lower circle of the pattern,

e.g., the $\phi$ of the top beam reaches $32°$ while the $\phi$ of the bottom beam is $27°$. To mimic the simulation setup of the 6-beam lidar in Fig. 4(a), we select 6 grid cells with different levels of opening angle (see Fig. 9), in which the central grid is always used. The mean half-cone opening angles of the 5 grid cells forming the circles are $12°$, $19°$ and $30°$, respectively. We estimate

the unfiltered radial velocity variance $\sigma^2_{v_r,\mathrm{unf}}$ using the Doppler radial velocity spectra collected in each selected grid cell. The Doppler spectra processing and usage are described in detail in Fu et al. (2022a).

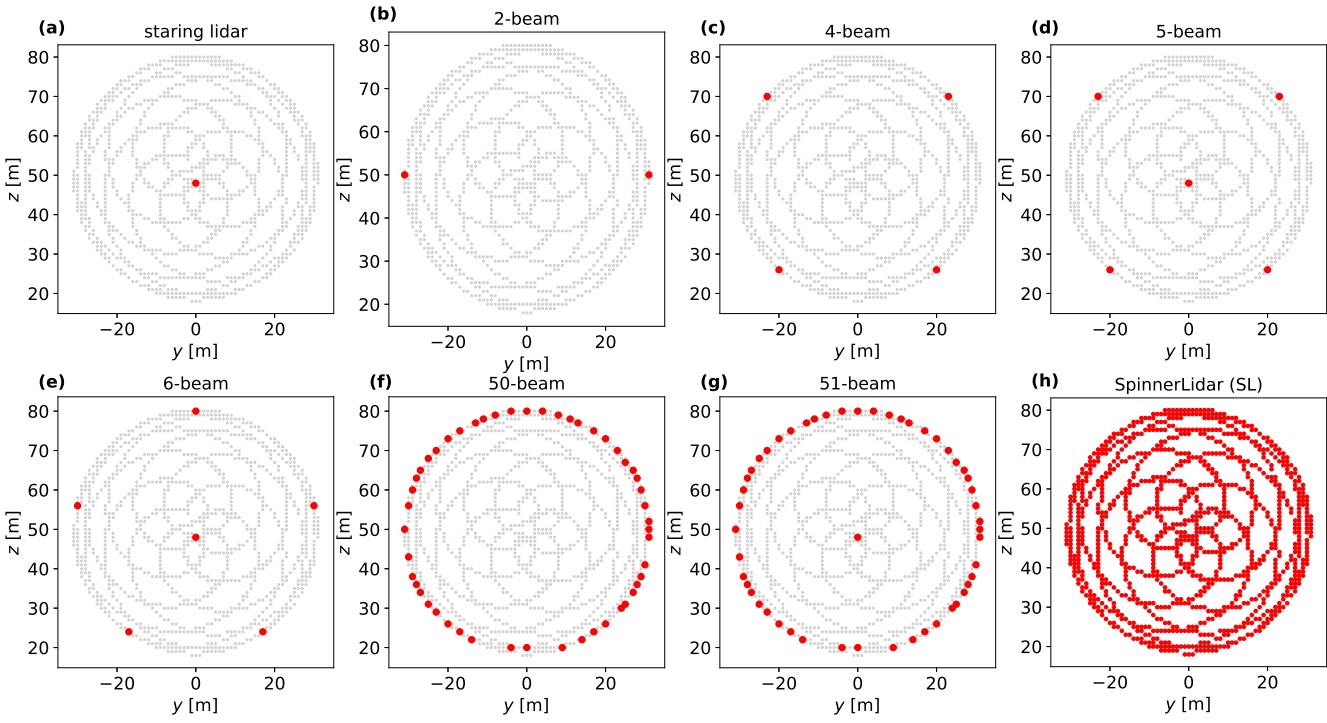

**Figure 8.** Selected lidar scanning patterns (in red) from the gridded SpinnerLidar scans (in light grey), which are at the focus distance of 62 m.

## 4 Results

In this section, we show comparisons of the Reynolds stresses computed from the considered lidars against those from the sonic anemometer at turbine hub height in bar plots. In the plots, markers correspond to the means of the estimations from 100 turbulence fields and the error bars are ± one standard deviation indicating the uncertainty of the estimation. The Reynolds stresses estimated from the measurements are normalized by the square of the mean along-wind velocity estimated by the lidar $U^2$ as we analyze a wide range of observed turbulence conditions. The mean wind velocity is computed by applying a least-square fit to the lidar radial velocities from all beams (Fu et al., 2022a). Results in Sections 4.1 and 4.2 neglect the lidars' probe volumes to study the influence of the number of beams. Nevertheless, for the CW lidar system, the probe volume increases with the square of the focus distance. Also, for pulsed lidar systems, the probe volume effect cannot be easily compensated since the Doppler spectra are usually not accessible. Therefore, the probe volumes are considered in Section 4.3 to show how different factors are altogether influencing the turbulence estimations.

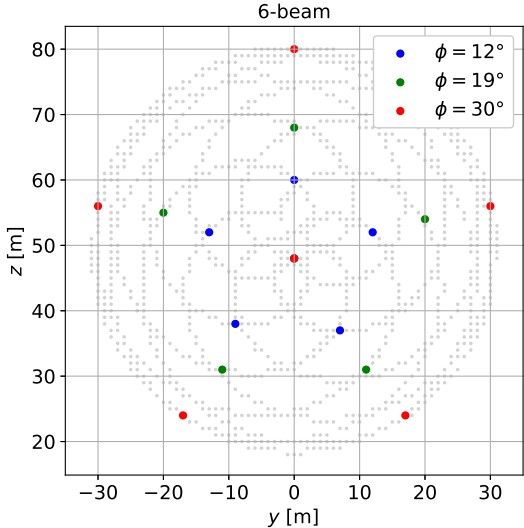

**Figure 9.** Selected grid cells for the 6-beam lidar with three different levels of the half-cone opening angle. The central grid coincides in the three cases. The gridded SpinnerLidar scans are shown in light grey.

## 4.1 Estimation of Reynolds stresses by multiple-beam lidars

We show in Fig. 10 the estimations of the six Reynolds stresses by the lidars, which have more than six beams and measure at a single plane, as well as those of the sonic anemometer. Results in Fig. 10(a) are from simulations that assume the lidars measure at the focus point only, i.e., no probe-volume averaging is accounted for. Results from both the simulations and the measurements show that the SpinnerLidar gives the best estimation for all six components. The results for the 6-beam and 51-beam lidars are very similar with larger errors and higher uncertainties than those of the SpinnerLidar. The 50-beam lidar can estimate the covariances accurately, while it shows large errors and uncertainties for $\langle v'v' \rangle$ and $\langle w'w' \rangle$; these are so noisy that some of them are out of the limit of the figure's axis. This is because the least-squares problem as formulated in Eq. (14) can lead to infinite solutions if we have only one opening angle $\phi$. By comparing the results from the 50- and 51-beam lidar, we can see that the addition of a central beam is very beneficial for the computation of the variances of the velocity components, because the central beam provides an additional opening angle to the 50-beam lidar making the matrix on the left side of Eq. (15) not singular. In principle, adding an extra beam in any different opening angle than the others in the 50-beam scanning pattern will improve the estimations. The central beam is the best option for improving the estimation of the $\langle u'u' \rangle$ since the beam aligns with the along-wind velocity component and can fully capture its variation when the probe volume is neglected.

Results in Fig. 10 (a) indicate that nacelle lidars are able to characterize inflow turbulence as accurate as the sonic anemometer with reasonable uncertainties, when the lidar has at least six beams and two different opening angles. We see the similar trends from the measurements shown in Fig. 10 (b). The unfiltered Reynolds stresses estimated from all lidar measurements

are generally close to those from the sonic anemometer but biased. What unexpected and rare are the negative values of $\langle v'v' \rangle$ and $\langle w'w' \rangle$ observed in some periods of the measurements, as shown and discussed in Fu et al. (2022a).

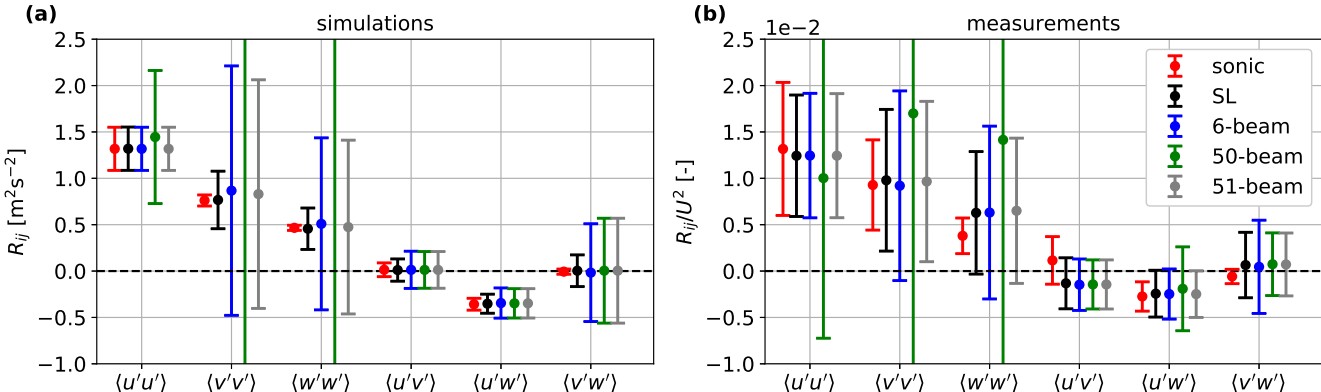

**Figure 10.** Reynolds stresses derived from the sonic anemometer and lidars, which have more than six beams and measure at a single distance. (a) simulated with 100 virtual wind fields. The lidars' probe volumes are neglected. (b) computed from the unfiltered radial velocity variance of the measurements. The markers are the means and the error bars are $\pm$ one standard deviation indicating the uncertainty of the estimation.

Figure 11 shows four of the Reynolds stresses retrieved from the 4- and 5-beam lidars. $\langle u'v' \rangle$ and $\langle v'w' \rangle$ are neglected in Eq. (15). In all cases, the determinants of the matrix in Eq. (15) are close to zero, which indicate that the 4- and 5-beam configurations cannot estimate these four Reynolds stresses accurately using the least-square procedure. Results from multiple-plane cases show that measuring at several planes with the same beam orientations does not aid much in the Reynolds stress reconstruction, as the determinant of the matrix in Eq. (15) does not change. For the 5-beam lidar, adding measurement planes only slightly reduces the uncertainty of the $\langle u'u' \rangle$ and $\langle u'w' \rangle$ components. This lack of sensitivity is partly due to Taylor's frozen hypothesis, as we do not account for evolution in the turbulence fields. We observe the same trend by comparing the estimation of these stresses from a 50-beam lidar measuring at a single and multiple planes (not shown here).

### 4.2 Estimation of the along-wind variance by all considered lidars

In case the nacelle lidar has fewer than six beams, not all six Reynolds stresses can be solved from Eq. (15). We focus our estimations on the along-wind variance and retrieve $\sigma_u^2$ from all considered lidars using the 'LSP-$\sigma_u^2$', 'LSP-isotropy' and 'LSP-IEC' methods, respectively, as introduced in Section 3.2. Results are shown in Fig. 12. All lidars are simulated to measure at a single plane (same as in Fig. 2) without accounting for the probe volume. Results from measurements are computed using the unfiltered radial velocity variances.

Both simulation and measurement results show, as a general trend, that lidar-derived $\sigma_u^2$ values are overestimated using the 'LSP-$\sigma_u^2$' method when compared to those from the sonic anemometer, while they are underestimated using the 'LSP-isotropy' method. The 'LSP-IEC' method gives the most accurate estimates among the three methods, as it assumes relations between the

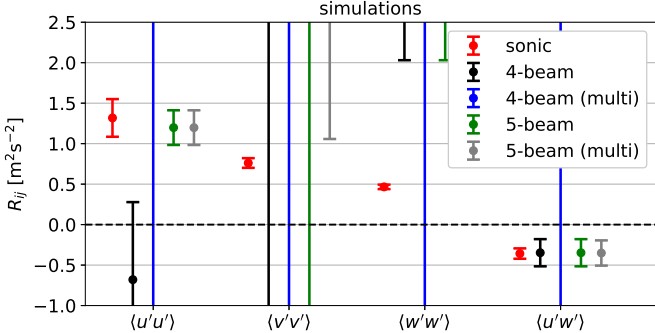

**Figure 11.** Reynolds stresses derived from the virtual sonic anemometer, the 4- and 5-beam lidars measuring at a single and multiple (multi) planes from 100 simulated wind fields. The lidars' probe volumes are neglected.

variances of the velocity components that might be close to those we can find within the atmospheric surface layer. The staring lidar performs like a sonic anemometer in our simulations as the beam is perfectly aligned with the along-wind component and the effect of lidar probe volume is not considered. Overall, all considered lidars are able to estimate $\sigma_u^2$ very well, despite of

295 their different number of beams.

Table 2 summarizes the relative errors of the means of lidar-derived estimates compared to the one from the sonic anemometer. A negative value indicates that the along-wind variance is underestimated and vice-versa. The results in the first row of the table are computed solving the full matrix of Eq. (15) (same as $\langle u'u' \rangle$ showed in Fig. 10, here denoted as 'LSP-6Re' method), from which we get perfect estimations of $\sigma_u^2$ using the 6- and the 51-beam lidars, and the SpinnerLidar without the effect of

300 the probe volume in the simulations. Furthermore, for lidars that have at least six beams and two different opening angles, the method 'LSP-6Re' is the best option to compute $\sigma_u^2$ among others, because it does not assume any relations between the six Reynolds stresses. While for lidars with fewer than six beams or only one opening angle, the 'LSP-6Re' does not work well and the 'LSP-IEC' gives the best estimation of $\sigma_u^2$. These results are aligned with one of the main findings in Fu et al. (2022a). In this work, the 'LSP-IEC' gives even smaller errors because we are able to compensate for the probe volume effect and use

the 'unfiltered' radial velocity variances. In addition, comparing the relative errors between the 4- and 5-beam lidars, and those between the 50- and 51-beam lidars, we find again that the addition of a central beam can sometimes improve the estimation of the along-wind variance.

### 4.3 Dependence of Reynolds stresses estimations on the opening angle, focus distances and the type of lidar

The results shown in this section include the averaging effect of the lidar probe volume. In Fig. 13, we analyze how the accuracy

and the uncertainty of the Reynolds stresses estimations change when increasing the half-cone opening angle $\phi$ for the 6-beam lidar. The simulation setup has been shown in Fig. 4(a). We compare these estimations with those from the sonic anemometer and the SpinnerLidar. The lidar probe volumes are modelled as in a CW system. Simulation and measurement results show that both the error and the uncertainty decrease as the opening angle increases. Specifically, the 6-beam lidar with $\phi = 45°$

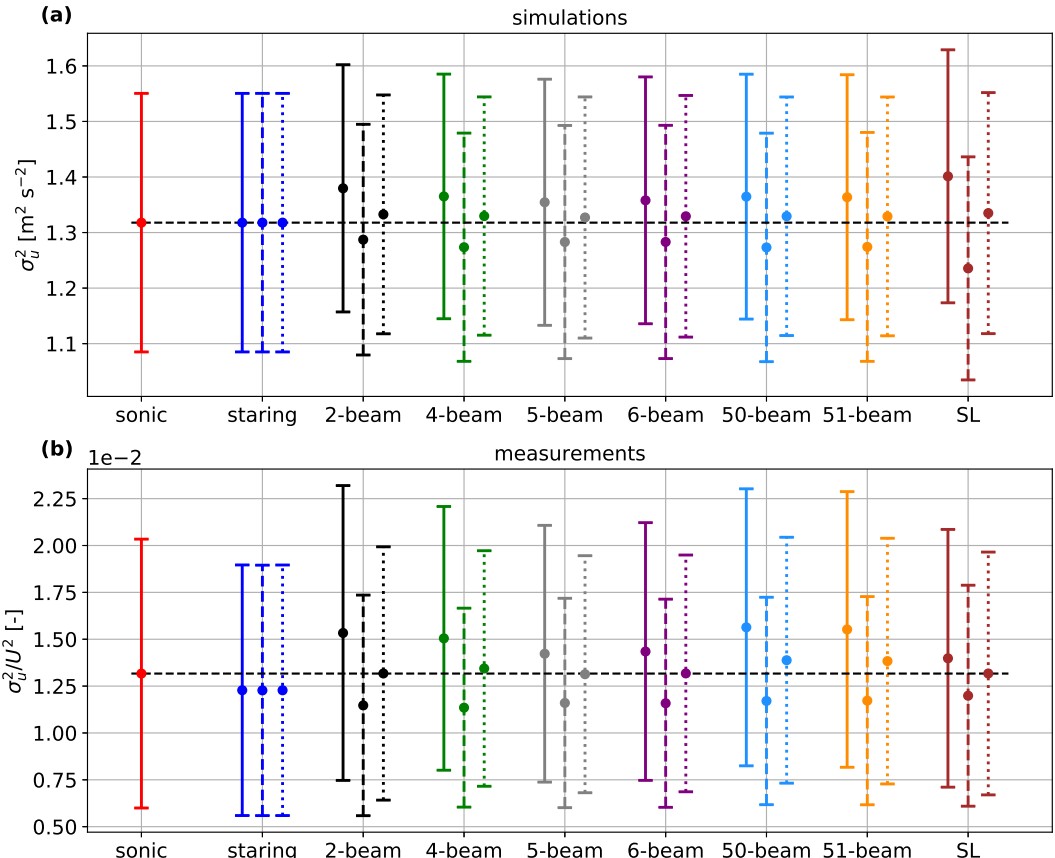

**Figure 12.** The along-wind variance derived from all considered lidars using the 'LSP-$\sigma_u^2$' method (solid lines), 'LSP-isotropy' method (dashed lines) and 'LSP-IEC' method (dotted lines). All lidars from simulations are assumed to have no probe volume and they measure at a single plane (Fig. 2).

in the simulations provides lower uncertainty than the SpinnerLidar despite having much fewer beams, as the SpinnerLidar's
maximum opening angle is $\phi = 30°$. We observe the same trend when simulating the probe volume with a 6-beam pulsed
system (not shown here). Possible reasons for the positive bias of the $v$- and $w$-variances seen from the simulation results are
discussed in Section 5.

We study the dependence of the Reynolds stresses estimations on the increasing focus distance $f_d$ for the 6-beam lidar
based on numerical simulations. The setup has been shown in Fig. 4(b). We assume the lidar systems to be continuous-wave
and pulsed, as shown in Fig. 14(a) and (b), respectively. All Reynolds stresses are computed using the centroid-derived radial
velocity variances. Therefore, the estimated variances are attenuated by the probe volume and in general smaller than those
from the sonic anemometer. For both types of lidar, we see that increasing the focus distance has negative effects on the
estimation of all Reynolds stresses. The uncertainty increases due to the random error on the variances of the radial velocity;

| | methods | staring | 2-beam | 4-beam | 5-beam | 6-beam | 50-beam | 51-beam | SL |
|---|---|---|---|---|---|---|---|---|---|
| | LSP-6Re | — | — | — | — | 0 | 9.7 | 0 | 0.1 |
| simulations | LSP-$\sigma_u^2$ | 0 | 4.7 | 3.6 | 2.8 | 3.0 | 3.6 | 3.5 | 6.3 |
| (without probe volume) | LSP-isotropy | 0 | −2.3 | −3.4 | −2.6 | −2.6 | −3.4 | −3.3 | −6.2 |
| | LSP-IEC | 0 | 1.1 | 0.9 | 0.7 | 0.9 | 0.9 | 0.9 | 1.3 |
| | LSP-6Re | — | — | — | — | −5.4 | −23.9 | −5.5 | −5.6 |
| measurements | LSP-$\sigma_u^2$ | −6.8 | 16.4 | 14.3 | 8.0 | 8.9 | 18.7 | 17.9 | 6.2 |
| (unfiltered variance) | LSP-isotropy | −6.8 | −12.9 | −13.8 | −11.9 | −12.0 | −11.1 | −11.0 | −9.0 |
| | LSP-IEC | −6.8 | 0 | 2.1 | −0.3 | 0.1 | 5.4 | 5.0 | 0 |

**Table 2.** Relative error [%] of the mean values of the lidar-derived along-wind variance to the one from the sonic anemometer. The lidars' probe volumes are neglected in the simulations. Results from the simulations are computed using measurements at a single plane (same set up as Fig. 2). A negative value indicates that the along-wind variance is underestimated and vice-versa.

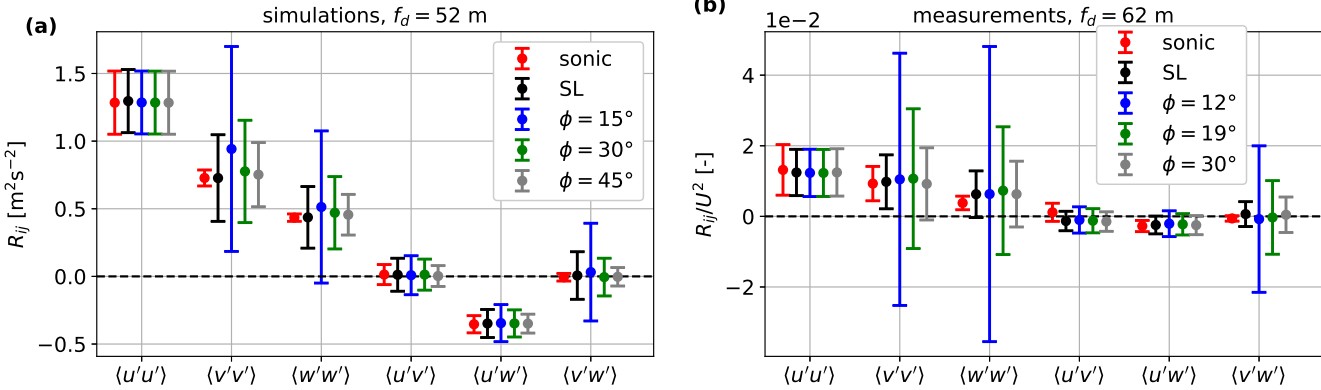

**Figure 13.** Dependence of the Reynolds stresses estimations on the increasing half-cone opening angle $\phi$ for the 6-beam lidar (single plane), the sonic anemometer and the SpinnerLidar ($\phi = 0$–$30°$). The probe volume in the simulations is assumed to be as in CW systems. All Reynolds stresses are computed using the unfiltered radial velocity variances.

they are less correlated when the lidar scans over a larger area. In the case of the CW system, the bias for the estimations increases with $f_d$ due to its growing probe volume, while the bias is almost constant for the pulsed system, as expected. For the closest focus distance $f_d = 52$ m, the bias of the estimations from the pulsed system is evidently larger than those from the CW system, where the later system gives accurate estimations of all Reynolds stresses. We perform the same analysis with the 51-beam lidar and observe the same trends (not shown here).

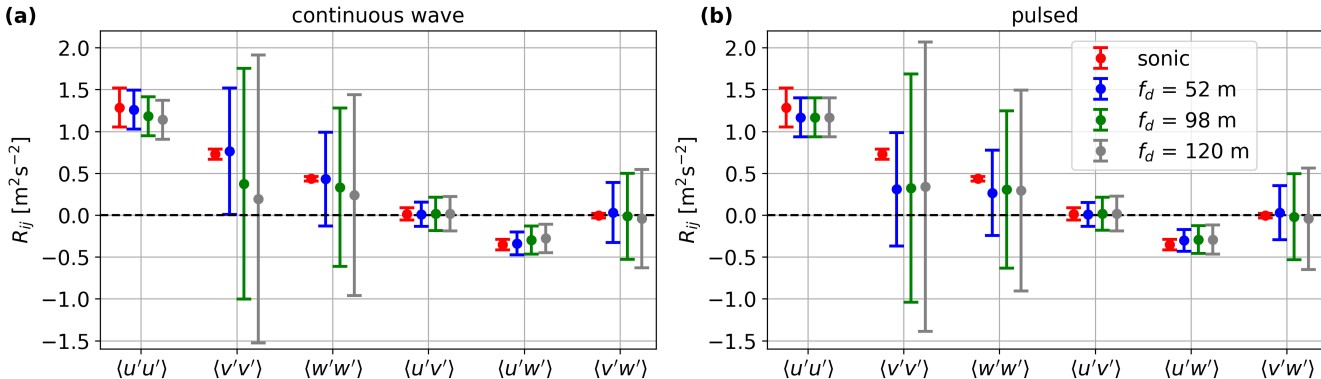

**Figure 14.** Dependence of the Reynolds stresses estimations on the increasing focus distance $f_d$ for the 6-beam lidar (single plane, $\phi = 15°$), compared to those from the sonic anemometer. The probe volume in the simulations are assumed to be as in (a) a CW system, and (b) a pulsed system. All Reynolds stresses are computed using the centroid-derived (filtered) radial velocity variances.

## 5 Discussion

Results shown in Fig. 13 are from simulations that consider the CW lidar probe volume to mimic the lidar's behavior in the reality. Then, the Doppler radial velocity spectra are used to compute the 'unfiltered' velocity variances for both simulations and measurements. Compared to the estimations from the sonic anemometer, we observe positive biases of the lidar-retrieved $v$- and $w$- variances. The biases decreas with increasing the half-cone opening angle $\phi$. The reason is that although the large matrix on the left side of Eq. (15) is not degenerate (i.e., its determinant is not zero) for a 6-beam lidar, the coefficients for $R_{vv}$ and $R_{ww}$ are very small (in the order of $10^{-3}$) for $\phi = 15°$; the equation system is only balanced by overestimating both terms $R_{vv}$ and $R_{ww}$. The coefficients are proportional to the value of the opening angle $\phi$, so they increase to $10^{-2}$ in the case of $\phi = 30°$, and to $10^{-1}$ in the case of $\phi = 45°$, which explains why the biases are reduced with larger opening angles. The positive biases for $R_{vv}$ and $R_{ww}$ are slightly more evident in the simulations with probe volume compared to the case in which the probe volume is neglected (see Fig. 10 (a)), because the simulated radial velocity variances are different in the two scenarios.

As shown in Fig. 13, increasing the lidar opening angle improves the accuracy and uncertainty of $R_{vv}$ and $R_{ww}$ estimations. The uncertainty of $\sigma_u^2$ is not much influenced if the lidar has a central beam that always aligns with the mean wind, e.g. the six-, 51-beam lidars, and the SpinnerLidar. For nacelle lidars without a central beam, enlarging the opening angle brings higher

uncertainty to $\sigma_u^2$ estimation, which is a key parameter for assessing wind turbine loads (IEC, 2019). Therefore, the optimum opening angle for turbulence estimations depends on which Reynolds stress is of interest. In addition, for control applications, the large opening angle is beneficial for measuring wind directions, but sacrifices the accuracy of rotor-effective wind speed and wind shear estimations (Simley et al., 2018). The optimum opening angle is also very much relevant to the turbine's size.

In this work, we characterize turbulence in front of a small wind turbine at $1D$ and $1.2D$ in the simulations and the field experiment, respectively. Taylor's frozen turbulence hypothesis (and homogeneity) is assumed throughout our numerical simulations, because the wind evolution is not very relevant to turbulence statistics, but more to the rotor-effective wind speed estimations (Chen et al., 2021). Mann et al. (2018) showed that turbulence is slightly affected by the stagnation in front of the wind turbine rotor as it goes through the induction zone. The change of the low-frequency wind variation is related to the thrust coefficient of the wind turbine, but the main turbulence statistics do not change. In addition, the yaw misalignment of the wind turbine is not considered in this work. A small yaw misalignment (below 20°) does not affect much $\sigma_u^2$ estimations but increases the uncertainty of $R_{vv}$ and $R_{ww}$ estimations. For modern wind turbines with very large rotor disks, the single-point turbulence statistics do not represent well the inflow turbulence affecting the wind turbine. The least-square procedure cannot be used to characterize the inhomogeneous inflow. New methodologies, e.g., constrained simulations (Dimitrov and Natarajan, 2017; Conti et al., 2021), are needed to reconstruct the inhomogeneous wind field.

We show from both simulations and measurements that all six Reynolds stress components can be estimated accurately when using a nacelle multi-beam lidar. Although the spectral turbulence model used here (the Mann model), which is the basis of our simulated turbulence fields, assumes two of these components to be zero, namely $\langle u'v' \rangle$ and $\langle v'w' \rangle$, the methods and techniques introduced in this work enable us to estimate all components accurately. This is advantageous for the study of atmospheric flow over complex terrain and, particularly, in offshore conditions, where turbulence measurements are scarce and expensive, and where we rely very much on models to assess the site conditions that impact wind turbines. These models often assume relations between the turbulence components and/or use parametrizations of stresses/fluxes that are invalid due to the nature of the flow phenomena and the interaction between the waves and the wind field. For example, surface stresses over long-lasting waves can be highly misaligned with the vertical gradient of the horizontal wind; most parametrizations of the air-sea interaction assume such an alignment to estimate momentum fluxes within the marine boundary layer. Offshore nacelle lidars can therefore help us understanding phenomena that are otherwise difficult to assess with traditional anemometry used for offshore wind power development.

## 6  Conclusion and Outlook

This study investigated the dependence of the Reynolds stresses estimations on different number of beams, half-cone opening angles, focus distances, single or multiple measurement planes, and different types of the Doppler wind nacelle lidars using both numerical simulations and measurements. The considered lidar scanning patterns included the staring lidar (single beam), the 2-, 4-, 5-, 6-, 50-, 51-beam lidars and the SpinnerLidar, which reports 400 radial velocities with one scan. We assumed a homogeneous inflow turbulence (both for the simulations and measurements) and the Taylor's frozen turbulence (for the simu-

lations). The lidar-retrieved turbulence estimations were compared with those from a sonic anemometer at turbine hub height. Analysis of both numerical simulations and measurements showed that to estimate all the six Reynolds stresses accurately, a nacelle lidar system with at least six beams is required. Also, one of the beams of this system should have a different opening angle. Adding one central beam improves the estimations of the velocity components' variances. Measuring at multiple planes with the same beam orientations only reduces the uncertainty but not the bias in the reconstruction, if Taylor's frozen turbulence hypothesis is applied. All considered lidars can estimate the along-wind variance accurately by using the least-squares procedure and the assumption that the relations of the velocity components' variances are as suggested in the IEC standard. Also, the Doppler radial velocity spectra are needed for the accurate estimations. For both CW and pulsed lidars, increasing the opening angle reduces both the error and uncertainty of the estimations, while increasing the focus distance has opposite effects. In short, from all tested scanning strategies, a 6-beam CW lidar measuring at a close distance with a large opening angle gives the best estimations of all Reynolds stresses. The optimum value of the opening angle depends on the Reynolds stress term of interest and also the wind turbines' size. Further studies or experiments are needed to study the best opening angle of the 6-beam lidar for different applications.

In this work, the single-point turbulence statistics are estimated using the least-square procedure, which assumes homogeneity over the lidar scanning area. Wind turbines nowadays are often operating inside a wind farm or have large spans over the swept area. The assumption of homogeneous turbulence can be violated under those conditions. Therefore, further studies on the optimized lidar scanning strategy for turbulence estimation should consider the inhomogeneity of the inflow. Additionally, the proposed nacelle lidar scanning strategies can be used to study the wind evolution, the spatial correlations of turbulence and estimate multi-point statistics, which better characterize the inflow that interacts with the turbine than the hub height ones. The wind field reconstruction of the inhomogeneous wind fields can benefit from constrained simulations, which incorporate lidar measurements into three-dimensional turbulence wind fields. Future works could also consider the non-Gaussianity of turbulence (Liu et al., 2010; Schottler et al., 2017) and the scale-dependent anisotropy of wind fluctuations (Syed et al., 2023).

*Data availability.*  Measurements from the SpinnerLidar are not publicly available due to a non-disclosure agreement between the authors and the provider of the data. Simulated nacelle lidar measurements are available upon requests.

*Author contributions.*  All authors participated in the conceptualization and design of the work. WF and AS performed numerical simulations of nacelle lidars without probe volume. AS extended the simulations with lidar probe volume. WF conducted the analysis of field measurements and drafted the manuscript. AP and JM supported the whole analysis. All authors reviewed and edited the manuscript.

*Competing interests.*  At least one of the (co-)authors is a member of the editorial board of Wind Energy Science. The authors have no other competing interests to declare.

*Acknowledgements.* The campaign was conducted as a part of the LIdar-assisted COntrol for RElidability IMprovement (LICOREIM) project at DTU Wind Energy. This study is funded by the European Union's Horizon 2020 research and innovation program under the Marie Sklodowska-Curie grant agreement No. 858358 (LIKE – Lidar Knowledge Europe, H2020-MSCA-ITN-2019).

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
