# Peer review of "Dependence of turbulence estimations on nacelle-lidar scanning strategies"

_Wind Energy Science, 2022_

## Referee Comment (RC1)

**Referee report on "Dependence of turbulence estimations on nacelle-lidar scanning strategies" by Wei et al.**

The manuscript assesses the impact of different scanning strategies of a nacelle-based Lidar on the accuracy of certain turbulence parameter estimations such as the components of the Reynolds stress tensor, which plays an important role in the characterization of inflow turbulence conditions as well as subsequent turbine load or power curve estimations. The results presented in the manuscript are based on atmospheric turbulence measurements from a SpinnerLidar and "numerical simulations" of inflow turbulence, in this case the well-known Mann wind field model. Model parameters as well as scanning strategies used in the synthetic wind field model were chosen in order to emulate inflow conditions and scanning strategies used in the SpinnerLidar measurement campaign. Furthermore, estimation of Reynolds stress tensor components by the various scanning strategies, which essentially differ in the number and spatial arrangement of points in the rotor plane (by systematically eliminating measurement points of the SpinnerLidar trajectories), are compared to measurements of a sonic anemometer at hub height (in case of the synthetic wind field, the time series extracted in the center of the box). Using a recently proposed least squares method that allows to determine Reynolds stresses from the actually measured radial velocity variances, it is found that at least six points in the rotor plane are needed for an accurate estimation of the six Reynolds stresses.

Overall, the manuscript addresses an important issue in wind energy, i.e., the accuracy of in-situ inflow turbulence measurements by Lidar and its limitations, especially in comparison to standard (but limited) anemometer measurements from a meteorological mast. It offers a rather comprehensive study of the advantages and disadvantages of different Lidar scanning strategies and might thus contribute to a better understanding of the statistical characterization of inflow wind fields and turbine loading. I especially appreciated the combination of theoretical modeling, by testing scanning strategies inside of a synthetic wind field, and the measurement campaign. Nonetheless, I find that certain aspects, e.g., the presentation of results from both the numerical as well as the measurement campaigns, could still be improved in a revised version of the manuscript.

Although, the manuscript is clearly structured (theoretical background, methodology containing "numerical simulations" and field measurements, and a results section), I found it rather difficult to assess its main takeaway messages. In particular, the results section is a mere juxtaposition of results (in the form of numerous plots) and their descriptions in the main text without providing further background or even highlighting their significance. E.g., the authors find in Fig. 9 that the introduction of an additional measurement point (from Fig. 7 (f) to (g)) significantly improves the estimation and uncertainty of the Reynolds stresses. This certainly seems to be a relevant result and would therefore deserve further discussion, which could be - to some extend - even speculative: Why is this additional center point leading to substantially better results? Does the additional center better grasp certain aspects of the turbulent fluctuations or only the shear profile? In the context of homogeneous isotropic turbulence, for instance, it was shown [R Stresing and J Peinke 2010 New J. Phys. **12** 103046] that the inclusion of a point of reference has a considerable influence on the statistics of turbulent fluctuations.

On a more general note, results shown in the current version of the manuscript are restricted to single point quantities such as the Reynolds stress tensor in Eq. (11) and its determination by different methods and the results are quite impressive. Many important statistical features of atmospheric turbulence, however, are actually contained in spatio-temporal correlations (correlation lengths, scale-dependent anisotropies, even the validity of Taylor's hypothesis used by the authors)

and even higher-order moments of the velocity field (intermittency or non-Gaussianity of turbulent fluctuations), see e.g., [Liu et al. Boundary-Layer Meteorol **134**— 243–255 (2010)] or [Böttcher, F., Barth, S. & Peinke, J. Stoch Environ Res Ris Assess **21**, 299–308 (2007)]. From the proposed measurement setup, which contains multiple spatial points, it should also be possible to at least determine spatial correlations between two points. I would appreciate the authors commenting on these issues, perhaps even in the outlook section of the manuscript.

Here, some further comments on the manuscript:

- Not every reader of Wind Energy Science might be familiar with the Reynolds decomposition of the velocity field $u_i(\mathbf{x}, t)$ into mean $\langle u_i(\mathbf{x}, t) \rangle$ and fluctuating parts $u_i'(\mathbf{x}, t)$. Please introduce the appropriate notation here; in the current version it is not clear that primed variables such as the ones in line 28 denote the fluctuating part of the velocity field.

- Line 249: Possible typo? Only off-diagonal stresses, such as $\langle u'v' \rangle$ or $\langle u'w' \rangle$ can become negative.

- Uncertainties of the 6- and 50-beam measurements in Fig. 9 extend beyond the range that is plotted, which the authors explain by a vanishing determinant of the matrix in Eq. (15). Is it correct that the values for $\langle u'u' \rangle$, $\langle v'v' \rangle$, and $\langle w'w' \rangle$ for the 50-beam simulations in Fig. 9 (a) also lies outside of the range although that the corresponding measurements in Fig. 9 (b) still lie within this range and generally exhibit higher uncertainties?

- In the results section, it is quite confusing that the authors first present their least squares method (Fig. 9) and then directly move on to the estimation of $\sigma_u$ with the three other methods discussed in Sec. 3.2. Please differentiate more clearly and explain that this is due to the fact that the least squares only works for more than six beams and that you have to rely on the methods discussed in line 155 otherwise.

- Concerning the model parameters in Sec. 3.3: Please explain the origin of the specific parameter values. As of now, it is not clear whether these model parameters were actually determined on the basis of the field measurements in Sec. 3.4 or not.

- The authors claim that accurate estimation of the Reynolds stresses requires at least six measurement points in the rotor plane. Is this due to the fact that the least square method is not working anymore or is it due to the fact that the measured radial velocity variances are not fully converged? Could the authors please be a bit more specific here.

---

## Author Response (AR1)

**Response to referee 1**

Dear referee,

Thank you for your general comments on our work, which we consider very important in helping us to improve the manuscript. Here is our response to each of your comments. Comments from the reviewer are reported in black and followed by our answers in blue.

Best regards,
The authors
* * *
**Main comments:**
(1). Although the manuscript is clearly structured (theoretical background, methodology containing "numerical simulations" and field measurements, and a results section), I found it rather difficult to assess its main takeaway messages. In particular, the results section is a mere juxtaposition of results (in the form of numerous plots) and their descriptions in the main text without providing further background or even highlighting their significance. E.g., the authors find in Fig. 9 that the introduction of an additional measurement point (from Fig. 7 (f) to (g)) significantly improves the estimation and uncertainty of the Reynolds stresses. This certainly seems to be a relevant result and would therefore deserve further discussion, which could be - to some extent - even speculative: Why is this additional center point leading to substantially better results? Does the additional center better grasp certain aspects of the turbulent fluctuations or only the shear profile? In the context of homogeneous isotropic turbulence, for instance, it was shown [R Stresing and J Peinke 2010 New J. Phys. 12 103046] that the inclusion of a point of reference has a considerable influence on the statistics of turbulent fluctuations.

*Adding a central beam to the 50-beam lidar is beneficial for computing the u, v, w velocity variances because the opening angle of the central beam is different from other beams in the circular scanning pattern. As we explained in Section 3.2 in the revised manuscript, "if the nacelle lidar has only one opening angle, some of the six equations from the least-square procedure will be linearly dependent, and we have fewer knowns than the unknowns, which leads to infinite solutions". Also, we added more explanation in Section 4.2: "In principle, adding an extra beam in any different opening angle than the others in the 50-beam scanning pattern will improve the estimations. The central beam is the best option for improving the estimation of $\langle u'u' \rangle$ since the beam aligns with the along-wind velocity component and can fully capture its variation when the probe volume is neglected."*

*To improve the results section, we added some background and highlighted the significance of the results in some places when needed.*

(2). On a more general note, results shown in the current version of the manuscript are restricted to single point quantities such as the Reynolds stress tensor in Eq. (11) and its determination by different methods and the results are quite impressive. Many important statistical features of atmospheric turbulence, however, are actually contained in spatio-temporal correlations (correlation lengths, scale-dependent anisotropies, even the validity of Taylor's hypothesis used by the authors) and even higher-order moments of the velocity field (intermittency or non-Gaussianity of turbulent fluctuations), see e.g., [Liu et al. Boundary-Layer Meteorol 134— 243–255 (2010)] or [Böttcher, F., Barth, S. & Peinke, J.

Stoch Environ Res Ris Assess 21, 299–308 (2007)]. From the proposed measurement setup, which contains multiple spatial points, it should also be possible to at least determine spatial correlations between two points. I would appreciate the authors commenting on these issues, perhaps even in the outlook section of the manuscript.

*We added some text in the outlook "In this work, the single-point turbulence statistics are estimated using the least-square procedure, which assumes homogeneity over the lidar scanning area. Wind turbines nowadays are often operating inside a wind farm or have large spans over the swept area. The assumption of homogeneous turbulence can be violated under those conditions. Therefore, further studies on the optimized lidar scanning strategy for turbulence estimation should consider the inhomogeneity of the inflow. Additionally, the proposed nacelle lidar scanning strategies can be used to study the wind evolutions, the spatial correlations of turbulence and estimate multi-point statistics, which better characterize the inflow that interacts with the turbine than the hub height ones. The wind field reconstruction of the inhomogeneous wind fields can benefit from constrained simulations, which incorporate lidar measurements into three-dimensional turbulence wind fields. Future works could also consider the non-Gaussianity of turbulence (Liu et al., 2010; Schottler et al., 2017) and the scale-dependent anisotropy of wind fluctuations (Syed et al., 2023.)"*

*Apart from the outlook, we also mentioned wind evolution at the end of Section 4.1 "For the 5-beam lidar, adding measurement planes only slightly reduces the uncertainty of the $\langle u'u'\rangle$ and $\langle u'w'\rangle$ components. This lack of sensitivity is partly due to Taylor's frozen hypothesis, as we do not account for evolution in the turbulence fields.", and in discussion "Taylor's frozen turbulence hypothesis (and homogeneity) is assumed throughout our numerical simulations, because the wind evolution is not very relevant to turbulence statistics, but more to the rotor-effective wind speed estimations (Chen et al., 2021)."*

**Further comments:**

(1). Not every reader of Wind Energy Science might be familiar with the Reynolds decomposition of the velocity field $u_i(x, t)$ into mean $\langle u_i(x, t)\rangle$ and fluctuating parts $u'_i(x, t)$. Please introduce the appropriate notation here; in the current version it is not clear that primed variables such as the ones in line 28 denote the fluctuating part of the velocity field.

*In the revised manuscript, we introduce the decomposition of the velocity field in the introduction and also at the beginning of section 2.1.*

(2). Line 249: Possible typo? Only off-diagonal stresses, such as $\langle u'v'\rangle$ or $\langle u'w'\rangle$ can become negative.

*Theoretically, only off-diagonal stresses can be negative. However, we observed negative $\langle v'v'\rangle$ and $\langle w'w'\rangle$ estimations from field measurements using the least-square procedure, which was unexpected. This occurs rarely and randomly. The same results were shown and discussed in Fu et al. (2022a). As the authors stated in Fu et al. (2022a), the possible reason is that the radial velocity variances of the SpinnerLidar highly decrease with the increasing opening angles. In this case, the turbulence homogeneity assumption is violated.*

(3). Uncertainties of the 6- and 50-beam measurements in Fig. 9 extend beyond the range that is plotted, which the authors explain by a vanishing determinant of the matrix in Eq. (15). Is it correct that the values for ⟨u'u'⟩, ⟨v'v'⟩, and ⟨w'w'⟩ for the 50-beam simulations in Fig. 9 (a) also lies outside of the range although that the corresponding measurements in Fig. 9 (b) still lie within this range and generally exhibit higher uncertainties?

*In Fig.10 (Fig.9 in the preprint), the velocity variances estimations from 50-beam lidar get out of the plotted range, both from numerical simulations and measurements. This is due to the linear dependency of Eq. (15), which resulted in infinite solutions when the opening angles of all beams are identical. So the estimations from 50-beam lidar are random.*

*The estimations from the 6-beam lidar do not get out of the plotted range, as the matrix on the left side of Eq. (15) is not singular. Its Reynolds stresses estimations are reliable because the 6-beam lidar meets the requirements to solve the six Reynolds stresses from the least-square procedure, i.e., has at least six beams and two different opening angles.*

(4). In the results section, it is quite confusing that the authors first present their least squares method (Fig. 9) and then directly move on to the estimation of $\sigma_u$ with the three other methods discussed in Sec. 3.2. Please differentiate more clearly and explain that this is due to the fact that the least squares only works for more than six beams and that you have to rely on the methods discussed in line 155 otherwise.

*In the revised version, we explained more about this in Section 3.2 and also at the beginning of Section 4.2.*

(5). Concerning the model parameters in Sec. 3.3: Please explain the origin of the specific parameter values. As of now, it is not clear whether these model parameters were actually determined on the basis of the field measurements in Sec. 3.4 or not.

*As explained in Section 3.3 in the revised manuscript, "The selected three parameters are adopted from Mann (1994) and characterize a neutral atmospheric stratification on a typical offshore site. The dissipation rate $\alpha\varepsilon^{2/3}$ is a scaling factor on the turbulence intensity". They were not determined based on the field measurements used in this work.*

(6). The authors claim that accurate estimation of the Reynolds stresses requires at least six measurement points in the rotor plane. Is this due to the fact that the least square method is not working anymore or is it due to the fact that the measured radial velocity variances are not fully converged? Could the authors please be a bit more specific here.

*We have modified section 3.2 to explain the reason more clearly. "To solve the six Reynolds stresses accurately from Eq. (15), two requirements of the lidar scanning pattern need to be fulfilled:*

*- the lidar has at least six beams or measures at six different locations within one full scan*

*- the lidar has at least two different opening angles.*

*If a lidar has less than six beams or only one opening angle and some of the six equations are linearly dependent, we have fewer knowns than unknowns in Eq. (15), which leads to infinite solutions. In that case, only the along-wind variance can be estimated well."*

**References:**

Chen, Y., Schlipf, D., and Cheng, P. W.: Parameterization of wind evolution using lidar, Wind Energy Science, 6, 61–91, https://doi.org/10.5194/wes-6-61-2021, 2021.

Liu, L., Hu, F., Cheng, X. L., and Song, L. L.: Probability density functions of velocity increments in the atmospheric boundary layer, Boundary-Layer Meteorology, 134, 243–255, https://doi.org/10.1007/S10546-009-9441-Z/METRICS, 2010.

Schottler, J., Reinke, N., Hölling, A., Whale, J., Peinke, J., and Hölling, M.: On the impact of non-Gaussian wind statistics on wind turbines – an experimental approach, Wind Energy Science, 2, 1–13, https://doi.org/10.5194/wes-2-1-2017, 2017.

Syed, A. H., Mann, J., Platis, A., and Bange, J.: Turbulence structures and entrainment length scales in large offshore wind farms, Wind Energy Science, 8, 125–139, https://doi.org/10.5194/WES-8-125-2023, 2023.

Fu, W., Peña, A., and Mann, J.: Turbulence statistics from three different nacelle lidars, Wind Energy Science, 7, 831–848, https://doi.org/10.5194/wes-7-831-2022, 2022a.

**Response to referee 2**

Dear referee,

Thank you for your general comments on our work, which we consider very important in helping us to improve the manuscript. Here is our response to each of your comments. Comments from the reviewer are reported in black and followed by our answers in blue.

Best regards,
The authors
* * *
**Main comments:**

(1). Probe Volume in simulations: It makes a lot of sense to me that in 4.1 and 4.2 the measurement volume is not considered, since the analysis with the measurement also uses the "unfiltered" radial velocities. Of course, the method from 3.1 could be also applied to simulated data, but this would add more complexity to the simulation and paper. However, in 4.3, volume averaging is applied. This then causes issues with the bias (first paragraph of Section 5) and also is a bit inconsistent. Therefore, I would propose the authors to consider following ideas (or any other which could solve the issue):

a) Add some lines in the discussion and in the beginning of Section 4 about that issue.

b) Apply the method from 3.1 also to the simulated measurements (if possible) from Section 4.3.

c) Ignore volume averaging in Section 4.3 as well, focus only on cw lidar.

For idea a and b, Section 4.3 could be renamed to "… lidar opening angle, focus distances, and lidar type". For idea c, you would need to remove the "lidar type dependency investigation" from the abstract etc. Personally, I tend to Option c, if not too much work, since the paper already provides a lot of information and the lidar type dependency investigation is maybe not so interesting as the rest.

*For Fig. 13 in the revised manuscript (Fig.12 in the preprint), we were doing option (b) to simulate and process lidar measurements in a way that is close to the one we applied to field measurements. The CW lidar probe volume was simulated and then the probe volume averaging effect was compensated by applying the method from Section 3, as mentioned in the caption of Fig. 13.*

*For Fig. 14 (Fig.13 in the preprint), we decided to do option (a). As explained at the beginning of Section 4, "Results in Sections 4.1 and 4.2 neglect the lidars' probe volumes to study the influence of the number of beams. Nevertheless, for CW lidar systems, the probe volume increases with the square of the focus distance. Also, for pulsed lidar systems, the probe volume effect cannot be easily compensated since the Doppler spectra are usually not accessible. Therefore, the probe volumes are considered in Section 4.3 to show how different factors are altogether influencing the turbulence estimations."*

*The positive bias of v and w variances estimations using a 6-beam lidar with 15 deg in Fig. 13 (revised manuscript) is slightly larger than those in Fig. 10, because the simulated radial velocity variances are different with and without considering the probe volume. As explained at the beginning of Section 5, the*

*biases are due to the small coefficients for $R_{vv}$ and $R_{ww}$ if the half-cone opening angle is small, and the matrix on the left side of Eq. (15) is ill-posed. The biases are overall consistent in the two figures.*

(2). The discussion section could be improved: The impact on the matrix from Equation 15 on the opening angle could be discussed in more detail. This might also help to give a more precise conclusion e.g. in the abstract, only a "large opening angle" is mentioned, but there should be an optimum, since e.g. for very large angles closer to 90 deg, the uncertainty for the u component should increase a lot. The first paragraph is also focusing on Figure 12 only and some discussion on the results from Section 4.1 and 4.2 could be added. Further, the second paragraph mentions the assumed frozen turbulence and homogeneity as well as the induction zone. Here a discussion of the impact of these effects could be added. The impact of other assumption/limitation of this study could be added, e.g. the limitation to small yaw misalignment angles.

*As we added in the discussion "As shown in Fig. 13, increasing the lidar opening angle improves the accuracy and uncertainty of $R_{vv}$ and $R_{ww}$ estimations. The uncertainty of $\sigma_u^2$ is not much influenced if the lidar has a central beam that always aligns with the mean wind, e.g. the six-, 51-beam lidars, and the SpinnerLidar. For nacelle lidars without a central beam, enlarging the opening angle brings higher uncertainty to $\sigma_u^2$ estimation, which is a key parameter for assessing wind turbine load. Therefore, the optimum opening angle for turbulence estimations depends on which Reynolds stress is of interest. In addition, for control applications, the large opening angle is beneficial for measuring wind directions, but sacrifices the accuracy of rotor-effective wind speed and wind shear estimations. The optimum opening angle is also very much relevant to the turbine's size.", and in conclusion "The optimum value of the opening angle depends on the Reynolds stress term of interest and also the wind turbines' size. Further studies or experiments are needed to study the best opening angle of the 6-beam lidar for different applications.".*

*We added some text to highlight the significance of the results in Sections 4.1 and 4.2.*

*We added some text in the discussion: "Taylor's frozen turbulence hypothesis (and homogeneity) is assumed throughout our numerical simulations, because the wind evolution is not very relevant to turbulence statistics, more to the rotor-effective wind speed estimations (Chen et al., 2021). Mann et al. (2018) showed that turbulence is slightly affected by the stagnation in front of the wind turbine rotor as it goes through the induction zone. The change of the low-frequency wind variation is related to the thrust coefficient of the wind turbine, but the main turbulence statistics do not change. In addition, the yaw misalignment of the wind turbine is not considered in this work. A small yaw misalignment (below 20°) does not affect much $\sigma_u^2$ estimations but increases the uncertainty of $R_{vv}$ and $R_{ww}$ estimations. For modern wind turbines with very large rotor disks, the single-point turbulence statistics are not a good representative of the inflow turbulence affecting the wind turbine. The least-square procedure cannot be used to characterize the inhomogeneous inflow. New methodologies, e.g., constrained simulations (Dimitrov and Natarajan, 2017;Conti et al., 2021), are needed to reconstruct the inhomogeneous wind field."*

**Specific comments on some minor details:**

(1). l82: you could introduce U.

*In the revised manuscript, we introduced U in L84.*

(2). 2.: A simple sketch for the angles would be helpful. You could also explain the coordinate system, which might be helpful to understand, why e.g. the opening angle is between the beam and the NEGATIVE x-axis.

*We added a sketch showing the definition of the coordinate system and the beam angles for nacelle lidar modelling, see Fig. 1 in the revised manuscript. The opening angle is between the beam and the negative x-axis in a downwind view, since the x-axis is along the mean wind direction and the beam is pointing towards the inflow.*

(3). Figure 2, 3, 6: the z label ("z [m]") is mirrored.

*Thanks for noticing that. We corrected the z label.*

(4). Equation 6: The error function usually uses \sqrt(\pi) in the denominator instead of \pi. And you could also mention that Erf is the error function.

*Yes, you are right. We corrected it. Thanks for noticing the typo.*

(5). L112: "frequencies … inside the volume are not considered except the dominant frequency detected by the centroid method": this is a bit inaccurate, since the range weighting function acts as a filter and damps high frequencies in the wind much more than lower frequencies, but is not perfect (i.e. considering only some frequencies).

*We agree. The sentence is changed to "Variances calculated from the centroid-derived radial velocities are attenuated by the lidar probe volume, which acts like a low-pass filter to the wind velocity fluctuations." In L115.*

(6). Equation (13): The partial derivative from Equation (12) "(…) ^2" should be "2*(…)*n_i*n_j". Maybe I miss something. But if not, you could add the 2. It does not make any difference, Equation (13) is correct without it, but it might be easier to reproduce.

*You are right. We added the 2 to Eq. (13) in the revised manuscript.*

(7). Equation (14) and (15): Similar to the previous comment, couldn't you simplify these equations by removing n_i and n_j, which is present on both sides? This would help with the discussion on the impact of this matrix. But maybe I totally misunderstood the complexity.

*No, we cannot cancel the $n_{i,j}$ that are present on both sides, because e.g. $\Sigma n_1 \sigma_{vr}^2$ is different as $n_1 \Sigma \sigma_{vr}^2$. The $n_1$ can be removed if they are the same for all beams, i.e., if the opening angles of all beams are identical.*

(8). l155: is a simple mean over all radial velocity components already providing already the variance in u, v, and w? There should be still some weighting with the cos^2, right? If so, this might be a bit misleading.

*Assuming turbulence is isotropic, i.e., $R_{uu} = R_{vv} = R_{ww} > 0$ and $R_{uv} = R_{uw} = R_{vw} = 0$, Eq. (15) can be written as $(\Sigma n_1^4 + \Sigma n_1^2 n_2^2 + \Sigma n_1^2 n_3^2) R_{uu} = \Sigma n_1^2 \sigma_{vr}^2$. If the lidar has only one opening angle, $n_1$ of all*

*beams are identical, the equation can be further simplified to $\Sigma(n_1^2 + n_2^2 + n_3^2)R_{uu} = \Sigma\sigma_{vr}^2$. From Eq.(3) we know that $n_1^2 + n_2^2 + n_3^2 = 1$, so $R_{uu} = \frac{\Sigma\sigma_{vr}^2}{N}$, where N is the number of beams.*

(9). Section 3.3: The scan rate of the lidar systems is not mentioned here. Did you use the time resolution of the wind field or did you use a certain scan rate (e.g. 200 Hz for the SpinnerLidar or 4 Hz for a typical pulsed lidar)?

*In the revised manuscript, we explained at the end of Section 3.3 that "The time lag between each measurement within a full scan is not considered but assumed that measurements are taken at the same time. In the numerical simulations neglecting lidar probe volume (see results in Sect. 4.1 and 4.2), the time resolution of the wind field is used as the lidar scan rate, i.e., lidars complete one full scan in dt = dx/U = 0.22 s. While in the simulations considering lidar probe volume (see results in Sect. 4.3), the lidars are assumed to finish a full scan in 2 s."*

(10). L183: here it is not clear to me, what do you mean with velocity bin. One would expect a discretization of the distance from -M to M. Also, the "bin^{-1}" for the unit of the resolution might not be necessary, since I think it is clear that the resolution is per bin.

*The Doppler radial velocity spectrum is a function of $v_r$, which is discretized into many bins. We changed the sentence to "The resolution of Doppler radial velocity spectrum is 0.1 m/s per velocity bin." in Section 3.3.*

(11). Table 1: the typical CW lidar has a beam radius of 28 mm, which would be 2.8 x 10^-2 m, not -4. I assume this is a typo, since for the 2.44 m mentioned in Section 3.4 one need 28 mm and 0.28 mm would be very small.

*You are right. We corrected the typo to $2.8*10^{-2}$ m. Thanks for noticing it.*

**Specific comments on some very minor details:**

(1). Table 1 and 2: I am not sure about the WES style, but in general it is a bit more common to use the caption on top rather below the table. Please check.

*We think this is the WES style in preprint. According to our experience, the captions will be on the top of the table in the final published version.*

(2). Figure 7, caption: you could avoid the line break between "62" and its unit "m", e.g., using the latex package siunitx.

*We corrected it in the revised manuscript.*

(3). l77: "Section 5" might be more consistent compared to "Sect. 5."

*We corrected it in the revised manuscript.*

(4.) When you have a list of numbers, e.g. l173 or l186, you could use white spaces after the comma.

*We corrected it in the revised manuscript.*

**References:**

Chen, Y., Schlipf, D., and Cheng, P. W.: Parameterization of wind evolution using lidar, Wind Energy Science, 6, 61–91, https://doi.org/10.5194/wes-6-61-2021, 2021.

Conti, D., Pettas, V., Dimitrov, N., and Peña, A.:Wind turbine load validation in wakes using wind field reconstruction techniques and nacelle lidar wind retrievals, Wind Energy Science, 6, 841–866, https://doi.org/10.5194/wes-6-841-2021, 2021.

Dimitrov, N. and Natarajan, A.: Application of simulated lidar scanning patterns to constrained Gaussian turbulence fields for load validation, Wind Energy, 20, 79–95, https://doi.org/10.1002/we.1992, 2017.

---

## Referee Report (RR1)

**Referee report on revised version of manuscript on "Dependence of turbulence estimations on nacelle-lidar scanning strategies" by Fu et al.**

I want to thank the authors for responding to my and the other referee's comments on their initial manuscript on "Dependence of turbulence estimations on nacelle-lidar scanning strategies" and for respecting them in a revised version of the manuscript. After assessing the provided material, I conclude that the manuscript improved considerably. The topic covered in the manuscript, i.e., the impact of Lidar measurement strategies on estimations of turbulence characteristics, is of great importance for the wind energy community. As described by the authors in the outlook section, it will be a task for the future to perform similar assessments of measurement strategies for higher-order statistical quantities, such as the kurtosis or the skewness of wind field fluctuations evaluated at two or even more points in the atmospheric flow. The manuscript will be an excellent contribution to the literature, and I suggest accepting it in its current form.